# Informative Point cloud Dataset Extraction for Classification via Gradient-based Points Moving

Wenxiao Zhang
University of Science and Technology
of China
Hefei, China
wenxxiao.zhang@gmail.com

Ziqi Wang
Wuhan University
Wuhan, China
ziqi.wangwang@gmail.com

Li Xu
Singapore University of Technology
and Design
Singapore, Singapore
li_xu@mymail.sutd.edu.sg

Xun Yang
MoE Key Laboratory of
Brain-inspired Intelligent Perception
and Cognition, University of Science
and Technology of China
Hefei, China
xyang21@ustc.edu.cn

Jun Liu*
Lancaster University
Lancaster, United Kingdom
j.liu81@lancaster.ac.uk

## ABSTRACT

Point cloud plays a significant role in recent learning-based vision tasks, which contain additional information about the physical space compared to 2D images. However, such a 3D data format also results in more expensive training costs to train a sophisticated network with large 3D datasets. Previous methods for point cloud compression focus on compacting the representation of each point cloud for better storage and transmission but ignore the improvements in training efficiency. In this paper, we introduce a new open problem in the point cloud field, named *point cloud condensation*: Can we condense a large point cloud dataset into a much smaller synthetic dataset while preserving the important information of the original large dataset? In other words, we explore the possibility of training a network on a smaller dataset of informative point clouds extracted from the original large dataset but maintaining similar network classification performance. Training on this small synthetic dataset could largely improve the training efficiency. To achieve this goal, we propose a two-stage approach to generate the synthetic dataset. We first introduce a nearest-feature-mean based strategy to initialize the synthetic dataset, and then formulate our goal as a parameter-matching problem, which we solve by introducing a gradient-matching strategy to iteratively refine the synthetic dataset. We conduct extensive experiments on various synthetic and real-scanned 3D object classification benchmarks, showing that our synthetic dataset could achieve almost the same performance with only 5% point clouds of ScanObjectNN dataset

compared to training with the full dataset. Codes are available at https://github.com/XLechter/PointCondensation.

## CCS CONCEPTS

• **Computing methodologies** → **Point-based models**.

## KEYWORDS

Point cloud Condensation, Point cloud, Shape Representation

**ACM Reference Format:**
Wenxiao Zhang, Ziqi Wang, Li Xu, Xun Yang, and Jun Liu. 2024. Informative Point cloud Dataset Extraction for Classification via Gradient-based Points Moving. In *Proceedings of the 32nd ACM International Conference on Multimedia (MM '24), October 28-November 1, 2024, Melbourne, VIC, AustraliaProceedings of the 32nd ACM International Conference on Multimedia (MM'24), October 28-November 1, 2024, Melbourne, Australia.* ACM, New York, NY, USA, 10 pages. https://doi.org/10.1145/3664647.3680767

## 1 INTRODUCTION

Point clouds have recently gained great popularity for representing objects in 3D vision [13, 16, 32, 36, 73, 74, 76]. Several 3D object datasets have been created, such as ModelNet [61], ShapeNet [9], and ScanObjectNN [53]. Also, networks specifically designed for processing point cloud have been proposed [20, 28, 30, 38, 41–43, 52, 56, 73]. Training networks that work on point clouds is often more computationally expensive than the 2D image counterpart, as point clouds have irregular structures where the traditional convolution operation cannot be directly applied[5, 6, 34, 35, 57, 62, 71, 72, 75]. Some attempts have been made to design a convolution operation that operates directly on points [28, 42, 52, 56]. However, they commonly need to search the nearest neighbor points to combine the local property, involving a greatly high computational cost. Though downsampling the point cloud to a smaller number of points is a solution, it will significantly reduce the testing accuracy.

When a new network architecture is developed, as the new network involves new combinations of different layers, it must be trained from scratch again on the training sets. This process can take a long time, especially if the training set is large or if the

---

*Corresponding author.

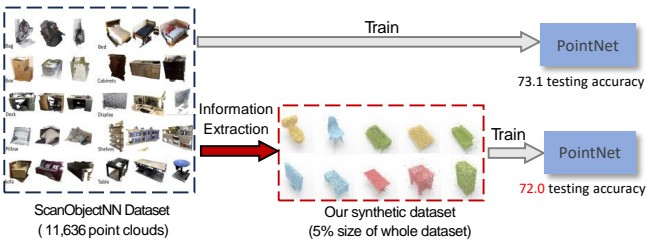

**Figure 1: We investigate the possibility of synthesizing a much smaller and informative training dataset from a large point cloud dataset. With our generated synthetic training set, the trained model could achieve similar testing accuracy on the ScanObjectNN classification benchmark with only 5% size of the whole dataset.**

network architecture is complex. Additionally, even though large volumes of point cloud data are requested by researchers to train neural networks, some data collectors may not want to make the data available to the general public for commercial reasons. For example, Shibuya et al. [51] have tried to transform point clouds into 3D line clouds to preserve point cloud data privacy.

Considering both the computation cost and data privacy, we investigate the feasibility of condensing a point cloud dataset into a compact set of representative synthetic point clouds, while preserving the majority of its information and minimizing performance degradation for classification task. Our objective is to create a synthetic dataset from a large dataset which should be significantly smaller in size compared to the original dataset, yet preserve the inherent characteristics of the dataset. Meanwhile, the generated synthetic dataset should be generalizable enough to train point cloud networks of different architectures. In this paper, we have explored that utilizing only a few synthetic point clouds generated with our method could result in plausible results when training the same model (Figure 1).

Similar tasks have been recently explored in 2D image areas, which are defined as image dataset condensation or distillation [55, 79, 81]. Most of these methods try to generate the synthetic image dataset to approximate the original training set via an iterative synthetic image optimization process. However, we find directly applying these methods to point cloud data is non-trivial where the generated synthetic point clouds are hard to converge to the optimal shapes by simply employing these methods. This challenge arises because 2D images are regular grids, allowing the optimization process to proceed effectively by simply altering pixel colors. In contrast, point clouds lack such regularity which allows points to move more freely. This complicates the optimization process because it becomes more difficult to determine the most effective optimization direction in 3D space.

We observe that the initialization of the synthetic point cloud dataset is a delicate factor which heavily impacts the synthetic set quality. In the 2D image field, most methods start with a synthetic dataset initialized with Gaussian noise or random samples, and often achieve satisfactory results after optimization. However, we find that due to the irregularity characteristic of point clouds, the initialization of the synthetic point cloud dataset is a delicate factor

that significantly influences the synthetic set convergence. In other words, a strategic and well-considered initialization is essential for obtaining the more informative synthetic point cloud dataset.

To address this challenge, we structure our objective by introducing a two-stage approach. The first stage is centered on devising an effective method for initializing the synthetic point cloud dataset. Specifically, we design a strategy based on the mean of the nearest point cloud feature, which entails initializing our synthetic dataset with a carefully selected subset of point clouds. In the second stage, we propose a parameter-matching strategy to optimize the synthetic point cloud through moving the position of points. Specifically, given the same network initialization, the parameters of the deep network trained on the original and synthetic datasets should ideally be very similar. This parameter matching issue could be solved through an iterative gradient matching strategy where the gradients for the backpropagation of two networks should be similar in each training iteration. We illustrate our method pipeline in Figure 2. Instead of leveraging point cloud generative models[1, 27, 33, 65] to generate synthetic point clouds, we directly regard the input synthetic point clouds as learnable parameters, and treat the network weights as a differentiable function to optimize the synthetic input. In each training iteration, we compute the difference between the gradients from the original and synthetic point clouds and finally the synthetic point clouds could be optimized via moving the points.

Comprehensive experimental evaluation of our approach on the various synthetic and real-scanned point cloud classification datasets shows the effectiveness of our method, demonstrating that we could train point cloud networks with small synthetic dataset to significantly fasten the training process with acceptable performance drop. To further demonstrate the effectiveness of our method, we further test our approach on practical applications such as continual learning in our experiments.

In summary, our contributions are as follows: **1)** We define a new open problem in the point cloud field, exploring the possibility of synthesizing a smaller training dataset from a large point cloud dataset for efficient training. **2)** We introduce a two-stage solution, in which we propose a nearest-feature-mean based initialization strategy, followed by a parameter-matching strategy through gradient-based points moving. **3)** Extensive experiments on various classification benchmarks show the effectiveness of our synthetic dataset, demonstrating we could train point cloud networks with a small synthetic dataset to significantly fasten the training process with acceptable performance drop. **4)** We further evaluate our approach through applications in point cloud continual learning, demonstrating the significance of the new task and the efficacy of our proposed method.

## 2 RELATED WORK

**Deep Learning on Point Cloud Analysis.** The field of point cloud processing has undergone a significant transformation with the advent of deep learning technology[29, 66–69]. Traditional methods that relied on hand-crafted local descriptors [10, 17, 31] have been gradually replaced by learning-based methods. Qi et al. proposed PointNet [41] as a solution to the challenge posed by the disordered nature of point clouds, incorporating shared MLP and max pooling layers. PointNet++ [42] was subsequently introduced to enhance feature discrimination through hierarchical local feature

extraction. Wang et al. introduced DGCNN [56], which utilized the distance between each point in the feature space instead of the Euclidean distance to determine the area surrounding each point for graph feature extraction. Other works have employed convolution neural networks to learn point cloud local characteristics [3, 14, 18–20, 26, 28, 30, 38, 43, 48, 49, 52, 63]. Despite the success achieved by these methods in analyzing point clouds, many of them require the calculation of point-wise distance information to aggregate local characteristics, resulting in a higher computational cost than the convolution of conventional 2D images.

**Generative models on point clouds.** Our work is also closely related to generative models for point clouds, including point cloud generative adversarial networks [1, 27, 33], optimal transport [70], and probabilistic-based models [24, 37, 65]. PointFlow[65], DPF-Net[22], and SoftFlow[25] utilize Normalizing Flows, while SetVAE [23] employs VAEs to generate sets of point clouds. ShapeGF[4] learns the distributions of gradient fields that constitute shape surfaces, and SP-GAN[27] utilizes a prior based on a spherical point cloud. However, the aim of these point cloud generation methods is to generate realistic-looking point clouds capable of fooling humans, while the objective of our work is to create informative training examples that can be effectively utilized for the training of deep neural networks.

**Dataset Compression & Condensation.** Recent studies [21, 58, 60, 64, 77, 80, 82, 83] have employed learning-based methods to compress point cloud data. Given the disorderly nature of point clouds, current methods [39, 46] usually use a sophisticated data structure, like an octree, to arrange the unprocessed point cloud data. However, these methods focus on improving storage and transmission efficiency rather than training efficiency. In the 2D image community, some researchers aim to choose a subset of the entire training dataset, referred to as a "coreset", which can be used to achieve good performance during training. Most of these methods [11, 47] progressively select important data points based on heuristic criteria. Recent works [8, 54, 55, 78, 79, 81] have been proposed to generate datasets that are not limited to the original dataset, but applying these works to point clouds is non-trivial due to the irregularity of point clouds as discussed in the introduction. To this end, we explore the possibility of synthesizing a smaller, informative training dataset from a larger point cloud set.

## 3 PROBLEM DEFINITION

Given a training set $\mathcal{T}$ consisting of $N$ point cloud and label pairs, $(x_i, y_i), i = 1, 2, \ldots N$, our objective is to generate a smaller set of synthetic point cloud and label pairs, $\mathcal{S} = (s_i, y_i)|i = 1, 2, \ldots M$, where $M(M \ll N)$ is much smaller than $N$. The ideal scenario would be a network trained on $\mathcal{S}$ converging faster and performing similarly to a network trained on $\mathcal{T}$.

The networks trained on $\mathcal{T}$ and $\mathcal{S}$ are denoted as $\phi_{\theta^\mathcal{T}}$ and $\phi_{\theta^\mathcal{S}}$, respectively, where $\theta^\mathcal{T}$ and $\theta^\mathcal{S}$ represent their respective parameters. The performance of each network can be defined as $\mathbb{E}_{x \sim P_\mathcal{D}}\left[\ell\left(\phi_{\theta^\mathcal{T}}(x), y\right)\right]$ and $\mathbb{E}_{x \sim P_\mathcal{D}}\left[\ell\left(\phi_{\theta^\mathcal{S}}(x), y\right)\right]$, where $P_\mathcal{D}$ represents the expected real data distribution and $\ell(., .)$ is the classification loss function. Our objective can be formulated as:

$$\mathbb{E}_{x \sim P_\mathcal{D}}\left[\ell\left(\phi_\theta^\mathcal{T}(x), y\right)\right] \simeq \mathbb{E}_{x \sim P_\mathcal{D}}\left[\ell\left(\phi_\theta^\mathcal{S}(x), y\right)\right]. \quad (1)$$

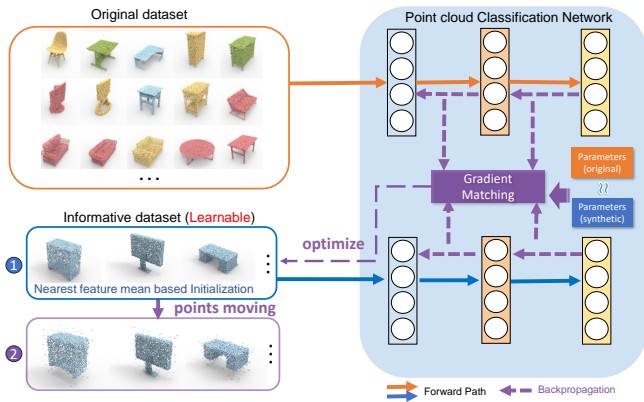

**Figure 2: Method Overview.** We learn an informative synthetic point cloud set which can get similar network parameters when a network is trained on it and the original dataset. We introduce a nearest feature mean based initialization strategy, and formulate our goal as an efficient gradient matching problem via points moving. The synthetic sets are regarded as learnable parameters and optimized iteratively.

In the experiments, the network performance is evaluated on the splited unseen testing set.

## 4 METHOD

To get the ideal $\mathcal{S}$ for a large point cloud dataset, we propose a two-stage solution. In the first stage, we introduce a nearest-feature-mean based initialization strategy, in which we initialize the synthetic point clouds with representative samples facilitating the following optimization process. In the second stage, we formulate our goal as a parameter-matching problem and further solve it by a gradient-matching strategy to iteratively optimize $\mathcal{S}$. We introduce our method details step-by-step in the following sections.

### 4.1 Nearest-feature-mean Based Initialization

As previously noted, the irregular nature of point clouds makes the initialization of $\mathcal{S}$ a sensitive factor in the subsequent optimization process. A good initialization can significantly enhance the convergence of $\mathcal{S}$ to get optimal performance. To define a good initial $\mathcal{S}$, inspired the heuristic sample selection method Herding [59], we consider $\mathcal{S}$ is well initialized if the initial $\mathcal{S}$ could approximate the overall distribution of the training set $\mathcal{T}$. To this end, we propose to initialize $\mathcal{S}$ by selecting the most representative samples in $\mathcal{T}$ based on the nearest feature mean.

Algorithm 1 describes our initialization strategy for $\mathcal{S}$. Specifically, $\mathcal{S}$ are initialized class by class. For the training set point clouds $\mathcal{T}_c$ belonging to class $c$, we extract the latent representation of each point cloud $x$ in $\mathcal{T}_c$ through a feature function $\phi_f : x \rightarrow \mathbb{R}^d$. For the feature function $\phi_f$, we train a point cloud classification network on $\mathcal{T}$, and use the last layer feature $f \in 1 \times D$ as the latent representation of each point cloud. Taking PointNet as an example, we use the feature after the max-pooling layer as the feature representation. Point clouds $s_1, \ldots, s_m$ are iteratively selected and accumulated until the desired total of $m$ is reached. At each

iteration step, a point cloud from the current dataset $\mathcal{T}_c$ is incorporated into $\mathcal{S}_c$. Specifically, the chosen sample is the one that most closely aligns the average feature vector of all samples across the entire training set. To this end, $\mathcal{S}_c$ could finally approximate the distribution of the original set $\mathcal{T}_c$.

---

**Algorithm 1** Synthetic Dataset Initialization Process
___

**Input:** point cloud dataset $\mathcal{T}_c = \{x_1, \ldots, x_n\}$ of class $c$
**Input:** $m$ target number of the synthetic dataset $\mathcal{S}_c$ of class $c$
**Require:** feature function $\phi_f : x \to \mathbb{R}^d$

$\quad \mu \leftarrow \frac{1}{n} \sum_{x \in \mathcal{T}_c} \phi_f(x)$   // current class feature mean
$\quad$ **for** $k = 1, \ldots, m$ **do**
$\qquad s_k \leftarrow \underset{x \in \mathcal{T}_c}{\arg\min} \left\| \mu - \frac{1}{k}[\phi_f(x) + \sum_{j=1}^{k-1} \phi_f(s_j)] \right\|$
$\quad$ **end for**
$\quad \mathcal{S}_c \leftarrow (s_1, \ldots, s_m)$
**Output:** Initialized synthetic dataset $\mathcal{S}_c$ of class $c$

---

## 4.2 Parameter Matching

To fulfill the objective outlined in Equation (1), it is crucial to not only achieve similar network output with minimal loss after training on $\mathcal{S}$ and $\mathcal{T}$, but also to obtain similar network parameters $\theta^{\mathcal{S}} \approx \theta^{\mathcal{T}}$, thus ensuring similar parameter optimization progress for improved generalization ability. Our goal can be expressed as:

$$\min_{\mathcal{S}} D\left(\theta^{\mathcal{S}}, \theta^{\mathcal{T}}\right), \tag{2}$$

$$\theta^{\mathcal{S}}(\mathcal{S}) = \arg\min_{\theta} \mathcal{L}^{\mathcal{S}}(\theta) \tag{3}$$

$$\theta^{\mathcal{T}}(\mathcal{T}) = \arg\min_{\theta} \mathcal{L}^{\mathcal{T}}(\theta) \tag{4}$$

where $D$ denotes the distance function between the parameters of the two networks, and $\mathcal{L}$ denotes the classification function.

In the process of parameter learning, the final trained $\theta^{\mathcal{T}}$ and the $\mathcal{S}$ is dependent on the initialization parameter $\theta_0$ which is sampled from an initialization parameter distribution $P_{\theta_0}$. Thus the above Equation 2, 3, 4 can be modified as:

$$\min_{\mathcal{S}} \mathrm{E}_{\theta_0 \sim P_{\theta_0}} \left[ D\left(\theta^{\mathcal{S}}(\theta_0), \theta^{\mathcal{T}}(\theta_0)\right) \right] \tag{5}$$

$$\theta^{\mathcal{S}}(\mathcal{S}) = \arg\min_{\theta} \mathcal{L}^{\mathcal{S}}(\theta(\theta_0)) \tag{6}$$

$$\theta^{\mathcal{T}}(\mathcal{T}) = \arg\min_{\theta} \mathcal{L}^{\mathcal{T}}(\theta(\theta_0)) \tag{7}$$

Original point cloud set $\mathcal{T}$ is time-invariant during training, therefore we can first get trained $\theta^{\mathcal{T}}$ in an offline manner, then $\theta^{\mathcal{S}}$ can be learned with $\theta^{\mathcal{T}}$ as the target.

The traditional approach to solve Equation 6, 7 involves implicit differentiation [44]. However, this can be complicated or expensive due to the formation of the derivative matrix. It becomes infeasible when the models are large. An alternative approach, back-optimization [15], is proposed where the network parameters ($\theta$) are partially optimized according to the loss function:

$$\theta^{\mathcal{S}}(\mathcal{S}) = Opt_{\theta}\left(\mathcal{L}^{\mathcal{S}}\left(\theta^{\mathcal{S}}\right), \mathcal{N}^{\mathcal{S}}\right) \tag{8}$$

$$\theta^{\mathcal{T}}(\mathcal{T}) = Opt_{\theta}\left(\mathcal{L}^{\mathcal{T}}\left(\theta^{\mathcal{T}}\right), \mathcal{N}^{\mathcal{T}}\right) \tag{9}$$

where $Opt$ means a specific optimization procedure with a fixed number of steps ($N$). However, using such a specific optimization procedure may not yield favorable results in our formulation. One reason is that the distance between the trained network parameters $\theta^{\mathcal{T}}$ and the initial network parameters $\theta^{\mathcal{S}}$ may be too large, and there may be multiple local minima in the loss reduction process, making it difficult to pass through.

## 4.3 Gradient Matching via Critial Points Moving

Instead of matching the parameters of $\theta^{\mathcal{S}}$ and trained $\theta^{\mathcal{T}}$, we match the updated $\theta_t^{\mathcal{S}}$ and $\theta_t^{\mathcal{T}}$ at each training iteration $t$ from the same initialization to overcome the problem of local minima traps. This converts the goal of parameter matching into smaller goals for each iteration which breaks down Equations (5, 6, 7) to:

$$\min_{\mathcal{S}} \mathrm{E}_{\theta_0 \sim P_{\theta_0}} \left[ \sum_{t=0}^{T-1} D\left(\theta_t^{\mathcal{S}}, \theta_t^{\mathcal{T}}\right) \right] \tag{10}$$

$$\theta_{t+1}^{\mathcal{S}}(\mathcal{S}) = Opt_{\theta}\left(\mathcal{L}^{\mathcal{S}}\left(\theta_t^{\mathcal{S}}\right), \mathcal{N}^{\mathcal{S}}\right) \tag{11}$$

$$\theta_{t+1}^{\mathcal{T}}(\mathcal{T}) = Opt_{\theta}\left(\mathcal{L}^{\mathcal{T}}\left(\theta_t^{\mathcal{T}}\right), \mathcal{N}^{\mathcal{T}}\right) \tag{12}$$

where $T$ denotes the total number of iterations. We hope that through this method of goal decomposition, we can train the network with the generated point cloud set $\mathcal{S}$ and original training set $\mathcal{T}$ to get the similar parameters after each iteration ($\theta_t^{\mathcal{S}} \approx \theta_t^{\mathcal{T}}$).

During the gradient descent optimization of iteration $t + 1$, the update rule of $Opt$ is:

$$\theta_{t+1}^{\mathcal{S}} \leftarrow \theta_t^{\mathcal{S}} - \eta_{\theta} \nabla_{\theta} \mathcal{L}^{\mathcal{S}}\left(\theta_t^{\mathcal{S}}\right) \tag{13}$$

$$\theta_{t+1}^{\mathcal{T}} \leftarrow \theta_t^{\mathcal{T}} - \eta_{\theta} \nabla_{\theta} \mathcal{L}^{\mathcal{T}}\left(\theta_t^{\mathcal{T}}\right) \tag{14}$$

where $\eta_{\theta}$ refers to the learning rate. For a network designed for point clouds, e.g., PointNet, the assumption is that at each iteration $t$, $\theta_t^{\mathcal{S}}$ and $\theta_t^{\mathcal{T}}$ will become equal. This allows the synthetic input $\mathcal{S}$ to be optimized by minimizing the difference between the mean gradients, based on gradient matching theory [45, 50, 81]:

$$\min_{\mathcal{S}} \mathrm{E}_{\theta_0 \sim P_{\theta_0}} \left[ \sum_{t=0}^{T-1} D\left(\nabla_{\theta} \mathcal{L}^{\mathcal{S}}(\theta_t), \nabla_{\theta} \mathcal{L}^{\mathcal{T}}(\theta_t)\right) \right] \tag{15}$$

$$\mathcal{L}^{\mathcal{S}}\left(\theta_t^{\mathcal{S}}\right) = \frac{1}{M} \sum_{(s,y) \in \mathcal{S}} \ell(\phi_{\theta_t}(s), y), \tag{16}$$

$$\mathcal{L}^{\mathcal{T}}\left(\theta_t^{\mathcal{T}}\right) = \frac{1}{N} \sum_{(x,y) \in \mathcal{T}} \ell(\phi_{\theta_t}(x), y), \tag{17}$$

where $D$ denotes the Mean Squared Error (MSE) between the two gradients at each layer.

Instead of directly minimizing the difference between the parameters $\theta_t^{\mathcal{S}}$ and $\theta_t^{\mathcal{T}}$, the gradient distance is used as a simplified alternative to calculate $\mathcal{S}$. This is because the gradient directly reflects the change in parameters, avoiding the need for recalculating the previous parameters $\theta_n$ ($n = 0, ..., n - 1$) when the parameters are not significantly different, which leads to faster convergence and lower memory consumption.

The gradient matching loss $D(\cdot; \cdot)$ in Equation 15 measures the distance between two gradients. According to the network layer type and width, the size of gradients is different in each layer. Here, we flatten the gradient vectors to $1 \times D_i$ where $D_i$ denotes the total gradient size in the $i$-th network layer. We compute the total layerwise gradient losses as:

$$d(\mathbf{G}^{\mathcal{S}}, \mathbf{G}^{\mathcal{T}}) = \sum_{i=1}^{\text{out}} \left( 1 - \frac{\mathbf{G_i^{\mathcal{S}}} \cdot \mathbf{G_i^{\mathcal{T}}}}{\left\| \mathbf{G_i^{\mathcal{S}}} \right\| \left\| \mathbf{G_i^{\mathcal{T}}} \right\|} \right) \qquad (18)$$

where $\mathbf{G_i^{\mathcal{S}}}$ and $\mathbf{G_i^{\mathcal{T}}}$ denote the gradients of $\mathcal{S}$ and $\mathcal{T}$ of the $i$-th layer, respectively. With gradient loss backpropagation, the points in $\mathcal{S}$ could move correspondingly to achieve similar gradients in each iteration. In our experiments, we find that those "critical points" move more significantly in the optimization process, which is discussed and illustrated in Sec. 6.2.

Additionally, the point cloud set $\mathcal{S}$ generated by our algorithm contains both point cloud data and the corresponding labels. Due to the complexity and learning difficulty, it is hard to optimize data and labels at the same time to get good results. Therefore, we fix a certain class label in a minibatch for generating the corresponding point cloud in a certain class.

## 5 TRAINING ALGORITHM

We illustrate our algorithm in Algorithm 2. We first choose a point cloud classification network backbone $\phi$, e.g., PointNet, and initialize the synthetic training set $\mathcal{S}$ with the introduced nearest-feature-mean strategy. The entire algorithm involves an outer loop for network parameter $\theta$ optimization and an inner loop for synthetic data $\mathcal{S}$ optimization. For the outer loop, we re-randomly initialize $\theta_0$ after $T$ iterations for $K$ times to increase the data generalization ability. In the inner loop, given the iteration $t$, we sample two training batches $B_c^{\mathcal{S}}$ and $B_c^{\mathcal{T}}$ from both the synthetic and original training set from the same class $c$. Then we compute the gradient matching loss between gradients $\nabla_{\theta} \mathcal{L}_c^{\mathcal{S}}(\theta_t)$ and $\nabla_{\theta} \mathcal{L}_c^{\mathcal{T}}(\theta_t)$ by applying gradient descent with learning rate $\eta_{\mathcal{S}}$. The gradient matching is performed class by class. After each iteration $t$, we optimize $\theta$ by computing the classification loss on the updated synthetic images $\mathcal{S}$. Though we could optimize $\mathcal{S}$ that are from multiple classes, we find that imitating the mean gradients of the data from a single class is easier compared to those of multiple classes.

## 6 EXPERIMENTS

**Evaluation Datasets.** We evaluate our proposed method on both synthetic and real-scanned classification datasets. **1) ModelNet.** ModelNet [61] is a widely used dataset for point cloud classification. Experiments are conducted on both ModelNet10 and ModelNet40. ModelNet10 contains 4,899 CAD models from 10 categories, with 3,991 shapes for training and 908 for testing. ModelNet40 contains 12,311 shapes from 40 categories, split into 9,843 for training and 2,468 for testing. **2) ShapeNet.** We also evaluate our methods on ShapeNetPart for classification, a subset of ShapeNet [9] with 16 categories containing 12,137 and 2,874 objects for training and testing, respectively. **3) ScanObjectNN.** ScanObjectNN [53] is a challenging real-scanned dataset containing 11,636 training objects and 2,814 testing objects that are categorized into 15 classes in the real world. We evaluated our method on ScanObjectNN without backgrounds.

**Evaluated Networks.** We use PointNet as the default classification backbone in all experiments if not specified. In the cross-architecture experiment, we test following 3D classifiers: PointNet [41], PointNet++ [42], DGCNN [56], PointNeXt [43], PointMLP [38], covering the pointwise-MLP-based, convolution-based, and graph-based methods.

**Experimental Settings.** We evaluate our method in three settings, generating a synthetic dataset containing one point cloud per category, 1% point clouds of the whole dataset, and 5% point clouds of the whole dataset. Each experiment involves two phases. First, we learn to synthesize a small synthetic set from a large real training set. Then we use the learned synthetic set to train randomly initialized neural networks and evaluate their performance on the real testing set. We use point clouds with 2048 points as inputs. We run each experiment 5 times and report the best performance of the learned synthetic set. The batch size $B_c^{\mathcal{T}}$ and $B_c^{\mathcal{S}}$ for sampling the original dataset and synthetic dataset are all set to 128 for PointNet and 32 for all other networks on a single A5000. We first use a learning rate of 0.0001 to optimize the synthetic point clouds for 1,000 iterations. After we get the synthetic dataset, we train new networks with a learning rate of 0.001 for 300 epochs with batch size 128.

**Baselines** We evaluate the performance of our method against baselines as follows:

- Random point clouds: It involves randomly selecting $M$ point clouds from each class in the training set.
- Herding: It uses the Herding algorithm [2, 7, 11, 59], which is the same as our initialization strategy to select representative samples without further optimization.
- K-Center: It uses the K-Center algorithm [47], another coreset approach, which selects multiple center samples and minimizes the largest distance between a data sample and its nearest center.
- DC [81] is a 2D dataset condensation approach. The core idea is to generate the synthetic dataset in a way that produces the same path of model updates as if training on the original dataset, which is similar to our gradient matching progress.

## 6.1 Performance comparison

In our experiments, we evaluated the performance of our proposed method by comparing it with various baseline methods, as depicted in Table 1. We employed PointNet as the default network backbone. Our results show that though some coreset methods, such as Herding and K-Center, outperformed random selection, our approach demonstrated a clear advantage over these selection methods as it was not restricted to a subset of the original dataset. Our approach outperforms those directly applying 2D dataset condensation methods, such as DC and DM, indicating the better convergence of our synthetic dataset. Our synthetic data achieved 73.5% accuracy using only 10 point clouds on the ModelNet10 dataset and 83.5% accuracy with only 5% of the point clouds. Our method also performed well on the ModelNet40 and ShapeNet datasets. On the real-world ScanObjectNN dataset, our method achieved 72.0 % accuracy using only 5% of the data, which is almost the same as the performance achieved using the entire dataset (73.1% accuracy).

---

**Algorithm 2** Synthesizing point cloud training dataset.

---

**Input:** Original Training dataset $\mathcal{T}$

**Initialization:** A point cloud classification network $\phi$, synthetic set $\mathcal{S}$ initialized with nearest-feature-mean strategy for $C$ classes, outer loop training iterations $K$, inner loop training iterations $T$, learning rate $\eta$.

1: **for** outer iteration k in K **do**
2:     Re-randomly initializing network parameters $\theta_0$
3:     **for** inner iteration t in T **do**
4:         **for** class c in C **do**
5:             Sample a minibatch pair $B_c^{\mathcal{T}} \sim \mathcal{T}$ and $B_c^{\mathcal{S}} \sim \mathcal{S}$   ▷ $B_c^{\mathcal{T}}$ and $B_c^{\mathcal{S}}$ are of the same class $c$.
6:             Compute the classification loss $\mathcal{L}_c^{\mathcal{T}} = \frac{1}{|B_c^{\mathcal{T}}|} \sum_{(x,y) \in B_c^{\mathcal{T}}} \ell\left(\phi_{\theta_t}(x), y\right)$ and $\mathcal{L}_c^{\mathcal{S}} = \frac{1}{|B_c^{\mathcal{S}}|} \sum_{(s,y) \in B_c^{\mathcal{S}}} \ell\left(\phi_{\theta_t}(s), y\right)$
7:             Update $\mathcal{S}_c \leftarrow \text{Opt } g_{\mathcal{S}}\left(D\left(\nabla_\theta \mathcal{L}_c^{\mathcal{S}}(\theta_t), \nabla_\theta \mathcal{L}_c^{\mathcal{T}}(\theta_t)\right), \eta_{\mathcal{S}}\right)$
8:         **end for**
9:         Update $\theta_{t+1} \leftarrow \text{Opt } \log_\theta\left(\mathcal{L}^{\mathcal{S}}(\theta_t), \eta_\theta\right)$ ▷ Use the whole $\mathcal{S}$
10:     **end for**
11: **end for**

**Output:** Synthetic training set $\mathcal{S}$

---

|  | Subset size | Ratio % | Training time (s) | Random | Herding | K-Center | DC | Ours | Whole Dataset |
|---|---|---|---|---|---|---|---|---|---|
| ModelNet10 | 10 | 0.25 | 1.8 | 39.3 | 69.6 | 69.6 | 71.5 | **73.5** | |
|  | 40 | 1 | 8.6 | 73.4 | 75.6 | 68.7 | 78.6 | **79.0** | 91.9 |
|  | 200 | 5 | 40.2 | 76.1 | 79.8 | 70.5 | 82.9 | **83.5** | |
| ModelNet40 | 40 | 0.4 | 7.0 | 35.3 | 45.3 | 45.3 | 47.5 | **49.5** | |
|  | 100 | 1 | 16.9 | 47.7 | 47.3 | 29.3 | 57.1 | **57.9** | 87.7 |
|  | 500 | 5 | 82.1 | 59.6 | 58.0 | 38.3 | 70.8 | **71.1** | |
| ShapeNet | 16 | 0.13 | 3.0 | 51.4 | 58.7 | 58.7 | 63.5 | **65.5** | |
|  | 122 | 1 | 22.9 | 76.9 | 77.5 | 72.5 | 80.9 | **81.1** | 97.1 |
|  | 607 | 5 | 109.2 | 89.3 | 84.5 | 80.4 | 86.7 | **88.4** | |
| ScanObjectNN | 15 | 0.12 | 3.0 | 23.9 | 31.0 | 31.0 | 32.6 | **34.7** | |
|  | 116 | 1 | 20.9 | 42.7 | 45.7 | 33.8 | 52.9 | **53.1** | 73.1 |
|  | 582 | 5 | 103.6 | 59.8 | 59.4 | 61.2 | 69.0 | **72.0** | |

**Table 1: The testing accuracy (%) of different methods on different datasets with PointNet for training and testing.**

| | ModelNet10 | | | | ModelNet40 | | | | ShapeNet | | | | ScanObjectNN | | | |
|---|---|---|---|---|---|---|---|---|---|---|---|---|---|---|---|---|
| Samples per class | 1 | 1% | 5% | whole set | 1 | 1% | 5% | whole set | 1 | 1% | 5% | whole set | 1 | 1% | 5% | whole set |
| Learning synthetic set | 1510s | 1816s | 2898s | - | 1789s | 2203s | 4782s | - | 1651s | 2325s | 5338s | - | 1630s | 2286s | 5270s | - |
| Training network | 5s | 21s | 105s | 2313s | 20s | 56s | 271s | 5320s | 8s | 67s | 312s | 5461s | 8s | 63s | 311s | 5295s |

**Table 2: Time consumption of learning the synthetic data and training PointNet with the synthetic data. The synthetic point cloud dataset only needs to be generated once.**

## 6.2 Visualization of synthetic point clouds

We visualize the learned synthetic point clouds of ModelNet10 and ModelNet40 in Figure 3. We find that the synthetic point clouds are visually recognizable, and we further conduct further analysis to explore why our generated synthetic point clouds yield better training performance.

In the visualization from Figure 3, we observe that the network changes some of the point positions, which leads us to consider whether our method is modifying some of the "critical points" of the original point clouds, which is discussed in PointNet and PointNet++ [41, 42]. Using PointNet as the backbone, we illustrate the critical points that activate the final max-pooling layer. We visualize the

synthetic point clouds and their corresponding critical points at different epochs in Figure 4. It can be seen that after optimization, the overall point clouds do not change much, but the critical points have been significantly altered, which verifies our hypothesis.

## 6.3 Synthetic set initialization strategies

In our default settings, we initialize our synthetic set with our proposed nearest-feature-mean based strategy. Here, we make a comparison via initializing the point clouds with random samples. In Table 3, we show the averaged results with 20 different random seeds. We could observe that we could get better performance with our initialized strategy on all datasets, indicating that the synthetic dataset $\mathcal{S}$ could converge better with our initialization.

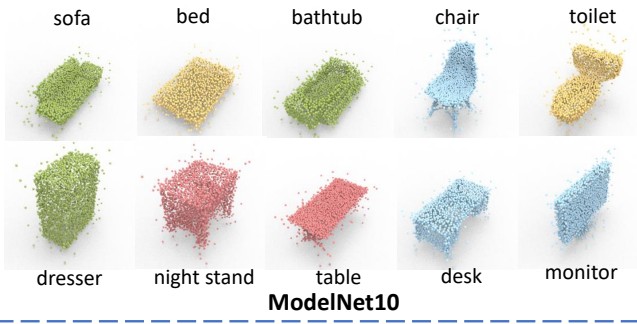

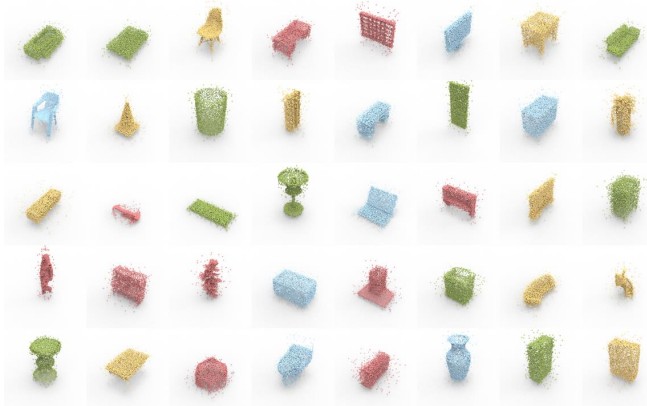

**Figure 3: Visualization of the learned synthetic data on ModelNet10 and ModelNet40 with one point cloud per class.**

| Samples | ModelNet10 | | | ModelNet40 | | | ShapeNet | | | ScanObjectNN | | |
|---|---|---|---|---|---|---|---|---|---|---|---|---|
| | 1 | %1 | 5% | 1 | %1 | 5% | 1 | %1 | 5% | 1 | %1 | 5% |
| Random | 71.5 | 80.0 | 83.1 | 47.5 | 57.9 | 71.1 | 63.5 | 80.9 | 88.4 | 31.7 | 52.9 | 69.2 |
| Ours | **73.5** | **79.0** | **83.5** | **50.5** | **60.9** | **72.1** | **65.5** | **81.1** | **90.4** | **34.2** | **53.1** | **72.0** |

**Table 3: Synthetic set initialized with random samples and our initialization strategy.**

## 6.4 Time consumption analysis

We give a time consumption analysis in Table 2 conducted on a single A5000. We report the time for learning the synthetic set with different numbers of point clouds per class, and the time for training PointNet with the synthetic set. Table 2 shows that the time for learning the synthetic set with 1 sample, 1% samples, and 5% samples per class for 1,000 iterations is less than training on the whole set for 300 epochs. With a smaller training set size, the training efficiency has been improved significantly. It is worth noting that **the synthetic point cloud dataset only needs to be generated once and then distributed to different users for efficient training without re-generation**.

## 6.5 Cross-architecture generalization

It is expected that the synthetic dataset will also be effective in training networks with different architectures. Thus we could generate a synthetic set using one backbone to gain generalization performance to different networks. We verify this through conducting two experiments that aim to evaluate the cross-architecture generalization.

To begin with, we analyze the synthetic datasets generated through three different PointNet architectures with different layer numbers and feature dimensions. Specifically, we employ three distinct PointNet configurations, namely the standard PointNet (which involves 5 shared Multi-Layer Perceptron (MLP) layers for feature extraction), the PointNet-small (which involves 3 shared MLP layers), and the PointNet-large (which involves 7 shared MLP layers). Specifically, we use shared MLP layers with (64, 128, 512) channels for PointNet-small, (64, 64, 64 128, 1024) channels for standard PointNet, and (64, 64, 64 128, 256, 512, 1024) channels for PointNet-large. Cross-testing is performed across these architectures, with results displayed in Table 4. It indicates that the generated synthetic datasets show commendable generalization performance on all of the PointNet architectures.

| Train/Test | PointNet | PointNet-small | PointNet-large |
|---|---|---|---|
| PointNet | 73.5 | 73.5 | 74.3 |
| PointNet-small | 71.2 | 70.6 | 71.9 |
| PointNet-large | 71.6 | 70.8 | 73.9 |

**Table 4: Cross-architecture testing accuracy (%) for one point cloud per class on ModelNet10.**

Nest, we generate synthetic datasets using a range of networks including PointNet, PointNet++, DGCNN, PointNeXt, PointMLP, and VoxelNet, all configured with default settings for classification. Cross-testing was performed across these networks and the results were evaluated on ModelNet10 with one point cloud per category, which are presented in Table 5. Results indicate that the synthetic dataset generated by PointNet produces the best performance on all the different architectures. **We hypothesize this is because all other baselines involve sorting and ranking operations (e.g., TopK operators in Pytorch) for performing local convolution, such as ball query (PointNet++) and group convolution (DGCNN) operators.** These kinds of operations are demonstrated as not gradient backpropagating friendly in [12]. Therefore, leveraging PointNet as the classification backbone, which is without sorting and ranking operations, could better optimize the synthetic dataset. Furthermore, the generated synthetic set using PointNet also performs well on other baselines, demonstrating that it could well represent the original set and is generalized with different architectures. Note that we set a small learning rate to 0.00001 for all baselines except PointNet, as we found that a larger learning rate will make the results collapes.

| Train/Test | PointNet | PointNet++ | DGCNN | PointNeXt | PointMLP | VoxelNet |
|---|---|---|---|---|---|---|
| PointNet | **73.5** | **68.3** | **71.2** | **72.3** | **73.9** | **70.2** |
| PointNet++ | 58.3 | 62.9 | 58.6 | 62.9 | 61.6 | 54.9 |
| DGCNN | 64.2 | 61.4 | 62.8 | 61.0 | 59.9 | 56.8 |
| PointNeXt | 61.3 | 63.4 | 60.8 | 63.8 | 60.6 | 58.7 |
| PointMLP | 62.3 | 60.4 | 61.4 | 59.6 | 61.0 | 59.0 |

**Table 5: Cross-architecture testing accuracy (%) with different networks for one point cloud per class on ModelNet10.**

## 6.6 Application

As outlined in our introduction, the synthesized compact point cloud dataset can be utilized in two key applications: 1) It can facilitate continual learning in point cloud field, when processing new,

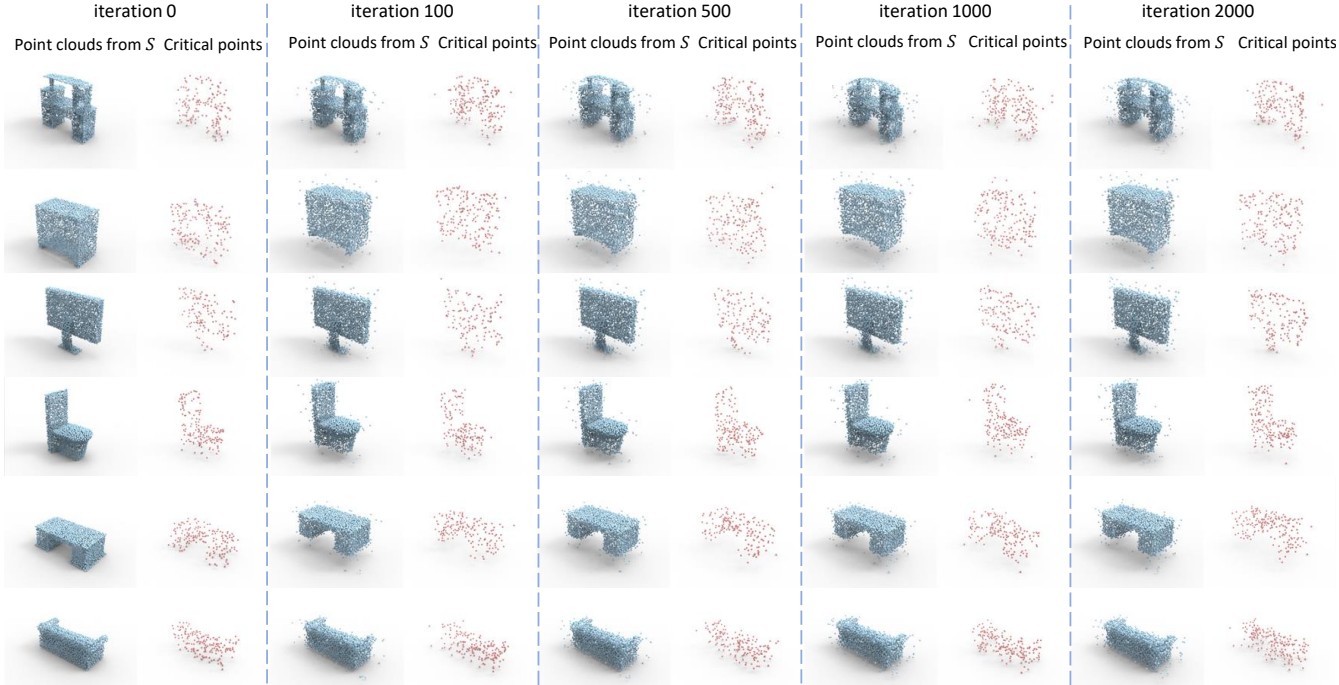

**Figure 4: Visualization moving process of the critical points which activate the max-pooling layer of PointNet.**

previously unseen classes of point cloud data. 2) Privacy concerns associated with the original data can be mitigated by sharing the generated datasets rather than the actual ones.

As outlined in our introduction, the synthesized compact point cloud dataset can be utilized in continual learning application. It can facilitate continual learning in point cloud field, when processing new, previously unseen classes of point cloud data.

**Point Cloud Continual Learning.** We adapted our method for continual learning to incrementally learn point clouds from new classes while preserving the performance on existing classes. We use the setting in [40] as our baseline, in which it progressively stores training samples in memory to maintain class balance. We replaced its Herding sample selection process by using our generated synthetic point cloud dataset. This includes generating synthetic point clouds from current classes and storing them in memory. We then assess the model performance in a task-incremental learning scenario using the ModelNet40 dataset of 40 classes.

Our method is benchmarked against Random selection, Herding, and DC, using a memory budget of 5 synthetic point clouds per class on the ModelNet40 dataset. We structure the learning into 5 and 10 steps, dividing the 40 classes into equal parts for each step. Utilizing the PointNet backbone, our approach consistently outperformed the others in both 5-step and 10-step learning scenarios which is shown in Figure 5. This indicates that our condensed dataset is more representative of the original dataset.

## 7 CONCLUSION AND FUTURE WORKS

The purpose of our paper is to explore the feasibility of training a point cloud classification network to obtain high accuracy using only a reduced number of synthetic point clouds derived from

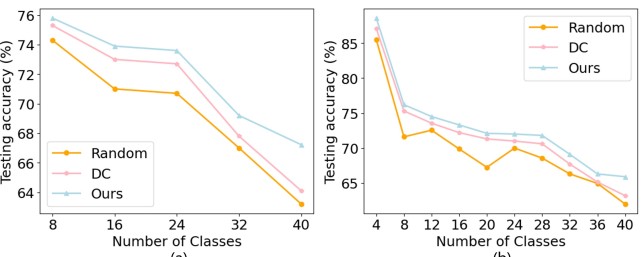

**Figure 5: (a) 5-step and (b) 10-step class-continual learning on ModelNet40.**

a large dataset. This objective is achieved by formulating it as a parameter-matching problem where a network can attain comparable network parameters after being trained on both the original and the generated synthetic point clouds. To resolve this challenge, a nearest-feature-mean based initialization strategy and effective iterative gradient matching approach are introduced. Experimental results demonstrate the effectiveness of the proposed method across various point cloud classification datasets and its application in point cloud continual learning and data privacy protection. In future work, besides the classification task, we aim to explore the application to more complex point cloud tasks.

## 8 ACKNOWLEDGEMENT

This research was supported by the National Natural Science Foundation of China (NSFC) under Grant U22A2094. This research was also supported by the advanced computing resources provided by the Supercomputing Center of the USTC.

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
