# OpenReview forum: "Informative Point cloud Dataset Extraction for Classification via Gradient-based Points Moving"
_acmmm.org/ACMMM/2024/Conference — MM2024 Poster_

### Official Review · Reviewer_HJAQ · 2024-05-21

**Rating:** 5
**Confidence:** 1

**Summary:**

In order to alleviate the high cost of data storage and training time for point cloud classification tasks, this paper innovatively proposes a gradient-based point movement method to extract small-scale data sets from large point cloud data sets. Furthermore, a nearest-feature mean based synthetic point cloud initialization method is introduced. The method proposed in this article can synthesize a new data set that is only 5% of the size of the original data set but can achieve comparable classification results.

**Strengths:**

1.The article raises a question worth pondering: Can a large data set be compressed into a small data set while maintaining performance on downstream tasks?

2.The article innovatively proposes a gradient-based point moving method. This method provides a new perspective for tasks such as data set extraction, generation, compression, and federated learning, that is, generating data with similar training effects based on gradient matching.

3.The writing is standardized and well structured.

**Limitations:**

1.Insufficient motivation. The gradient-based point movement method is the main argument of the article, however, the motivation for it is not fully demonstrated. The initialization method also does not have enough preparation.

2.The experimental results are not sufficiently discussed. It is recommended to supplement each experiment with a corresponding discussion.

3.From the visualization results, the initialization method almost dominates the characteristics of the generated data set. Therefore, more comparisons and discussions of different initialization methods are expected to be reported to illustrate the impact of initialization methods on point movement methods.

4.Few references in the past two years.

**Suitability:**

3

---

### Official Review · Reviewer_b9RV · 2024-05-23

**Rating:** 3
**Confidence:** 4

**Summary:**

This paper proposes a method for distilling point cloud datasets to generate a more informative distilled dataset, referred to as the Informative Point Cloud Dataset. To achieve this objective, the authors propose a two-stage approach to generate the Informative point cloud synthetic dataset. Initially, a strategy based on the nearest feature mean is introduced to initialize the synthetic dataset. Subsequently, a gradient matching strategy is employed to iteratively refine these initial point clouds.

**Strengths:**

1. The authors' idea of introducing dataset distillation from existing image datasets into point clouds is quite interesting.
2. The paper is well-organized and easy to read, with ample references provided.

**Limitations:**

1. Although the concept of dataset distillation proposed by the authors is quite interesting, it seems that this work primarily involves generating a smaller point cloud guided by the original complete point cloud through classification loss. This smaller point cloud is then used for training or testing a new network. This approach appears to be quite similar to existing learnable point cloud downsampling methods. The authors should clarify the differences between their proposed dataset distillation and current learnable downsampling approaches.

2. Distilling denser point clouds into sparser, meaningful point clouds is a significant task as it can promote the practical application of dense point clouds. However, the experiments in the paper are overly simplistic.
 1) Simple Datasets: The authors only analyzed relatively simple object point cloud datasets such as ModelNet, ScanObjectNN, and ShapeNet, which inherently contain a small number of points. Dataset distillation experiments would be more meaningful on more realistic and complex datasets, where an object consists of hundreds of thousands or millions of points.
 2) Simple Tasks: The authors only experimented with classification tasks. In fact, classification is a simple task where a few key points are often sufficient to achieve the same accuracy as a point cloud with a large number of points. This does not adequately demonstrate the true significance of dataset distillation. It is recommended that the authors conduct more extensive experiments on more complex analytical or generative tasks.

For these reasons, the authors' simplistic experimental results do not sufficiently support their overall motivation and ideas. Therefore, I am inclined to reject this paper and encourage the authors to conduct more meaningful and comprehensive experiments to support their story.

**Suitability:**

2

---

### Official Review · Reviewer_T67Y · 2024-05-27

**Rating:** 5
**Confidence:** 3

**Summary:**

This paper focuses on a very interesting task, i.e., information point cloud extraction task, where the authors tend to extract a small but informative sub-point set to achieve an algorithm with acceptable performance. Specifically, the authors develop a two-stage point removing algorithm, including nearest-feature-mean based initialization for small point set and gradient-matching strategy based parameter-matching solver. The empirical experiments show the effectiveness of the developed algorithm on popular ModelNet, ShapeNet, and ScanObjectNN datasets.

**Strengths:**

1. This paper studies a very interesting problem, namely, how to sample a small-scale informative point cloud set from a large-scale raw point cloud dataset to maintain performance without degradation.

**Limitations:**

1. How to choose the feature function in the nearest-feature-mean based initialization stage? The authors seem to choose an already trained PointNet encoder as the feature selection function. What dataset is the PointNet encoder trained on?

2. The number of the informative sub-point set is manually selected according to the prior knowledge. The authors should add additional ablation experiments to verify the impact of different number of point sets.

3. The strategy of extracting the subset of informative point clouds is similar to the idea of information bottleneck. I would like to see the author give more theoretical explanation from the perspective of information bottleneck. In addition, additional citations on information bottleneck should be added.

4. Is it possible to extend the proposed method to other point cloud analysis tasks, such as point cloud registration?

**Suitability:**

2

---

### Meta-Review · Area_Chair_z5UZ · 2024-06-28

**Recommendation:** Accept (Poster)
**Confidence:** 3

**Metareview:**

Initially the paper received 2 positive and 1 negative ratings. After the rebuttal, two reviewers raised their ratings, and all reviewers reached a concensus on accepting this work. After carefully reading both the reviews and the paper, the AC agrees with the reviewers' recommendations, since this submission hold merits in introducing an interesting task and approach for 3D point cloud learning. Although this work only conducts experiments on fundamental 3D classification task using a few baselines, the AC expects it to be a pioneering work that can inspire more following trials.

---

### Meta-Review · Senior_Area_Chairs · 2024-07-10

**Recommendation:** Accept (Poster)
**Confidence:** 4

**Metareview:**

This paper received mixed ratings initially. After rebuttal, all the reviewers tend to accept the paper. SAC and AC agree with the reviewers and recommend acceptance of the paper.